# Organizational Capital: A Resource for Changing and Performing in Public Administrations

**Barbara Barbieri** [1], **Ilaria Buonomo** [2], **Maria Luisa Farnese** [3,*] **and Paula Benevene** [2]

1   Department of Political and Social Sciences, University of Cagliari, 09123 Sardinia, Italy; barbara.barbieri@unica.it
2   Department of Human Sciences, LUMSA University, 00193 Roma, Italy; i.buonomo1@lumsa.it (I.B.); benevene@lumsa.it (P.B.)
3   Department of Psychology, Sapienza University of Rome, 00185 Roma, Italy
*   Correspondence: marialuisa.farnese@uniroma1.it

**Abstract:** The aim of this study was to deepen our knowledge about the role played by organizational capital (OC) among public administration (PA) agencies. A questionnaire was administered to a gender-balanced convenience sample of 270 workers of Italian PAs. First, confirmatory factor analysis was performed in order to examine the measurement model. Second, a SEM model was performed, confirming that OC was both directly and indirectly positively related to performance, through the mediation of innovation. OC was also positively related to innovation through the mediation of clarity about change. Overall, the results supported the hypothesized model, providing initial evidence on the pivotal role OC plays, and especially for PA agencies, on organizational innovation and performance. The limits and practical implications of these results are discussed.

**Keywords:** organizational capital; public administration; clarity related to change; innovation; organizational performance

## 1. Organizational Capital: A Resource for Changing and Performing in Public Administrations

The conceptualization of intellectual capital was born and developed in the private sector. However, a number of studies confirmed the relevance of addressing the knowledge of intellectual capital and its management through the intellectual capital theory among public administration (PA) agencies (see for instance: [1–4]). This theory assumes that knowledge is the most important resource for any organization to survive and grow, postulating that the intellectual capital paradigm is at the basis of the management of organizational knowledge. Intellectual capital is the sum of the intangible assets which, turned into organizational knowledge, help to create added value, sustainability, and a competitive advantage. It is expressed in its three dimensions: human capital (employees' knowledge and experience which are reversed in the organization), relational capital (the organization's relationship with employees, investors, customers, suppliers, and stakeholders) and organizational capital (OC) (knowledge resources rooted at the organizational level, such as processes, databases, policies, culture, vision, mission, and value statement).

The knowledge possessed by individual members of an organization needs an effective structure, systems, and modus operandi to achieve its full potential and transform the individual knowledge of employees into organizational knowledge [5–7]. More precisely, OC may develop the most appropriate context for procedures, processes and tools able to sustain the sharing of experience, information, and rough data, which are processed through the organization. In turn, this promotes the development of organizational value and competitive advantage [8].

The intellectual capital theory can serve PA agencies because not only do they have financial targets, but also different non-financial objectives. These include, but are not

limited to, as offering quality services, identifying new needs and shaping new services, and reaching all possible users of their services. Moreover, the outputs of PAs are mainly services, which are in their nature intangible and cannot be measured and assessed only through the lens of financial factors [9]. Moreover, the knowledge generated by their human resources and management is a fundamental intangible asset, and the academic and managerial literature has shown that these intangibles can be successfully managed through the intellectual capital lens (see for instance [10–13]). Finally, PA agencies are facing an ever-growing request for innovation as well as the improvement of their performance, in order to tackle the challenges posed by their changing environment. This is especially the case nowadays, when they are addressing a severe shortage of financial and human resources while at the same time dealing with new social emergencies.

As far as Italy is concerned, it must be noted that PA agencies are still following the "bureaucratic" model, that is "service oriented" instead of being "customer oriented" [14]. At the same time, the government is pushing for a change in the organizational culture of PA agencies, therefore adopting criteria of evaluation and assessment against strategic objectives, aimed at increasing the level of quality, effectiveness, efficiency, and cost-effectiveness of their activities. The Italian PA is an interesting forum to observe as it shows a similar pattern of change already present in the rest of Europe [15], namely a strong pressure for both innovation and higher performance on PA agencies.

## 2. OC, Innovation and Performance

A number of studies grounded on the Intellectual Capital theory have shown that there is a link between OC, on the one hand, and the generation of innovation capacity and the performance of firms, on the other [16,17].

OC is generally regarded as knowledge embedded in organizational routines, structures, systems, culture, values, and processes, thus constituting an important resource for organizational effectiveness [8,18,19]. In the public sector, OC concerns the set of knowledge of explicit or implicit nature, both formal and non-formal, that structures and develops the organizational activity in efficient and effective ways [20].

According to the intellectual capital approach, OC incorporates the knowledge developed by human capital into the organization through its structures [21], thus acting as a drive for the generation of innovation capability, as well as higher organizational performance. OC has been described as "what remains in the company when employees go home at night" [22]. In fact, unlike the human capital (which is "stuck" to the employees [23]), and the relational capital (which is external to the organization), OC is easier to manage and control by the organization itself, since it is the only capital that is completely owned by the organization [24]. OC is then the supportive infrastructure of knowledge [25].

It is a fact that most of the study carried out on OC and its effects on innovative capacity and performance have been mostly addressed in the private sector, while the PA has been rather neglected [26–28]. The aim of this study was to deepen the knowledge about the role played by OC among PA agencies in generating a better organizational performance and innovation capacity. To the best of the authors' knowledge, no previous study addressing this question among PA agencies has been carried out. This point needs to be further explored, and not only for academic purposes—namely the need to deepen the knowledge on how OC operates within specific contexts such as PA agencies—but also from the managerial point of view, since PA managers frequently have limited power to maneuver financial or human resources; nonetheless, they can have much more room to decide on the management of their intangible organizational resources.

## 3. The Psychosocial Approach to Organizational Capital

For the purposes of this study, the psychosocial factors of the working environment were used to describe and analyze the content of OC and to observe the extent to which they play a role on performance and innovation. Psychosocial factors of the workplace refer to certain aspects of the job and work environment, such as involvement in decision-

making processes, perceptions of efficient services to respond to users' needs, effective internal communication tools and processes, enhancement of teamwork, reflexive spaces dedicated to learning and knowledge development, error management as a source of improvement, and the perception of management support. These deal with the perspective of employees' experiences of their jobs and the organization as a whole. Psychosocial factors can be seen as intangible assets of the organization, since they are intangible in their nature and at the same time, they have been proven to generate performance, as well as competitive advantage and innovation. Furthermore, they can capture the relationship between the employees and their specific working context, thus offering the opportunity to deepen the understanding of the dynamics acting in the process of value creation through OC in a given type of organization [25]. This means observing how OC plays an actual transformative role, generating performance and innovation whilst considering the context-specific nature of OC [29] and thus focusing the research question on "how does OC work in a PA?".

Specifically, in a context where OC is recognized as a key resource for the organization, the use of intangible assets enables the organization to transfer the knowledge between both individuals and groups within the organization, paving the way to better organizational performance [30–32]. Furthermore, the knowledge integration process at the organizational level allows to recognize what is learned and to incorporate new information and skills in routines that will guide the future organizational actions [33,34]. Valuable knowledge, once captured and codified, can systematically be transmitted and disseminated, and other individuals can use it in new contexts [35] thus fostering the organizational capability to innovate [36].

The importance of knowledge and intangible assets has been widely recognized, including for PA agencies; however, there are still few studies that explore the relationship between OC and positive organizational outcomes in these organizations. Drawing on the well-established literature on OC and organizational performance and innovation, we assumed that this relationship could also be extended to the PA context. Therefore, two hypotheses were developed:

**Hypothesis (H1).** *OC is positively related to organizational performance.*

**Hypothesis (H2).** *OC is positively related to organizational innovation.*

Previous studies have suggested that organizations, by adopting new ideas or behaviors leading to the revision of their processes or products/services, increase their future performance, thus supporting a positive innovation–performance relationship [37]. The environmental challenges and the introduction of new management practices require higher capability to innovate, including in the public sector. Nonetheless, mission and resource constraints, limited competitiveness, and different criteria for performance evaluation due to the nature of their services, make the innovation–performance relationship among PA agencies an issue deserving further exploration [38].

Thus, the following hypothesis was also developed:

**Hypothesis (H3).** *Organizational innovation is positively related to performance (H3a) and mediates the OC–performance relationship (H3b).*

## 4. Organizational Capital, Clarity about Change and Organizational Outcomes

In order to generate proper innovation, the knowledge generated by the OC needs to be interiorized. That is, employees must not only be aware of the need for innovation, but also have to understand how future changes will affect their work and the life of the organization, so that they can effectively endorse the innovation [39]. In fact, according to Nonaka [40], knowledge must be internalized by members of the same organization, becoming a shared intangible resource of the whole organization, in order to effectively close and afterward re-start the virtuous spiral where tacit knowledge is turned into explicit

knowledge. Thus, it can be assumed that, when employees feel they have an opportunity to know about the implemented changes, in the framework of open communication with their managers, they will show higher alignment with the organizational objectives.

The need for a change in PAs is a process often introduced by law. For instance, in Italy, the national context of the present study, two important law reforms have concerned the PA in the last 15 years: the Brunetta Reform D.lgs. n.150/2009, focused on optimizing the performance of public employees, and the Madia Reform, Legislative Decree 74/2017. They both focused on the efficiency and simplification of the functioning of PAs, aiming to reduce the burden of bureaucracy and pushing towards a more innovative management.

Italian PA agencies are facing a strong request for change and innovation, which was enacted by the law and did not start as the result of the organizational strategy, thus it is pivotal for PA management to be able to communicate to their employees the innovation they are going to deal with, as well as to make them able to share the goals of the changes that are going to come. These are key factors to gaining employees' alignment and commitment to the changes and the innovations that are due to take place.

For this reason, we took into consideration the effect of the clarity about changes on positive outcome implementation (i.e., innovation and performance).

Many studies have highlighted how management can successfully support the organizational changes and innovation, including in the public sector [41–44]. Thus, delivering a clear vision of the change goals and endorsing an effective communication on them may represent crucial factors for the management of PA agencies, boosting employees' active participation in the change processes [45].

Moreover, the extent to which employees have a common understanding of their organization's vision is a prominent predictor of innovation. Indeed, to make lasting change and to incorporate the new policies or practices into their daily routines, employees must clearly understand how the proposed changes will work in daily practice [46]. Therefore, the following hypothesis was developed:

**Hypothesis (H4).** *OC is positively related to employees' clarity about change.*

The recognition of having precise and detailed information on one's own role within the change process as well as the perception of being an active actor of these changes are to be regarded as key elements for instigating innovation and organizational performance.

Clearly stated goals help employees to perceive them as attainable, to align their behaviors with them, investing additional efforts, enhancing their commitment, and their sense of responsibility [39].

Clarity about changes appears to encourage the fulfilling of reciprocal expectations between the organization and its employees, which in turn positively affects employees' engagement in innovative behaviors [47]. Organizational changes, when accompanied and supported by the management, can enhance organizational development and innovation [48].

Therefore, the following two hypotheses were developed:

**Hypothesis (H5).** *Clarity about change is positively related to organizational innovation.*

**Hypothesis (H6).** *Innovation mediates the clarity about the change–performance relationship.*

Figure 1 shows the theoretical model we tested.

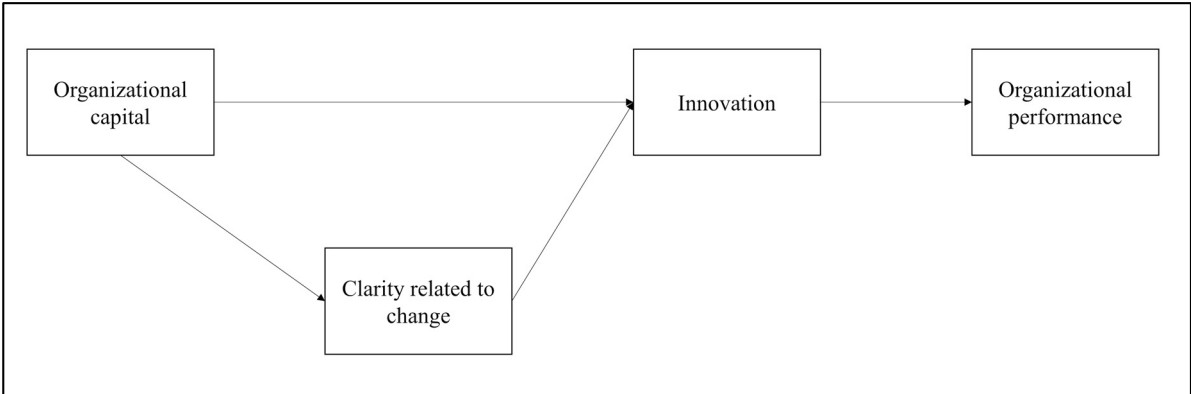

**Figure 1.** Theoretical model.

## 5. Method

*Context, Participants and Data Collection Procedure*

In this study, we focused our attention on PA agencies.

The deep social, economic and technological transformations of the last thirty years, accelerated by the economic and financial crisis of 2008, have highlighted the need for the PA sector to continuously redefine and improve its strategies, programs, and interventions. More specifically, since the 1990s, Italian PAs have been subject to several reforms, inspired by the international New Public Management reform doctrine [49] and requiring new ways of operating. Italy might then represent an interesting case to explore the relationship between OC and innovation and performance.

Our study reached a group of 270 employees, working in different Italian PA agencies. More specifically, 30% of them worked in PA agencies aimed at promoting public health, 18% worked in regional and local administrations, 13% worked in municipalities, 8% worked in schools or universities as administrative staff, and 33% worked in other types of PA agencies (e.g., related to environment, labor policies, revenue offices). Participants were aged from 26 to 66 years (men = 47.85, SD = 8.96; females = 34.8%). Of these, 65.2% ($n$ = 176) had at least a university degree, whilst 34.8% ($n$ = 94) had a high school or a lower degree; 32.6% ($n$ = 88) had management roles, and 62.9% ($n$ = 168) had more than 10 years of work experience.

Data were collected through anonymous questionnaires, which were administered through a Google drive questionnaire. The questionnaire administration was conducted through the snowball technique, which involved Bachelor's degree students who voluntarily took part in the data-collecting phase after a training session. To ensure the heterogeneity of the sample, each research assistant approached between 10 and 30 employees from different PA agencies. Participants voluntarily participated in the study and did not receive any reward. All participants were informed of the anonymity and confidentiality of the survey. The compilation was made in the period between 1 July 2020 and 31 August 2020.

## 6. Materials

*6.1. Organizational Capital*

Drawing on the OC literature, in our study, we operationalized some of the OC components suggested by Dumay [50] and Bontis and colleagues [51], more pertinent to the PA context, considering the psychosocial factors influencing the relationship between the work environment and individuals' behaviors. This choice was done taking into consideration that: (*a*) OC in fact depends on the type of organization itself and the context of PA agencies presents specific characteristics that distinguish these organizations, primarily their specific and substantial function of responding to the needs of the community through the provision of services for citizens; (*b*) previous literature has explored this intangible resource mainly in its structural components that are objectively measurable (for example patents, certifications, productivity or profitability), but less in its components with a dynamic

procedural nature (for example, one's know-how, information sharing, and involvement of employees in the decision-making process).

Thus, we measured OC with seven items assessing psychosocial aspects of organizational capital, so that the higher the score, the higher the perceived level of OC among the employees. The items were measured with a 5-point Likert scale (1 = totally disagree, 5 = totally agree). A sample item is: "When working in teams, we reflect together to decide how to adapt our intervention to reach the best possible outcome". The scale showed an excellent internal consistency (Cronbach's alpha = 0.934). Table 1 shows items and their psychometric characteristics.

**Table 1.** Organizational capital items and their psychometric characteristics.

| Item | Mean | Standard Deviation | Scale Mean if Item Deleted | Scale Variance if Item Deleted | Cronbach's Alpha if Item Deleted |
|---|---|---|---|---|---|
| We take quick and effective decisions *(Si prendono decisioni rapide ed efficaci)* | 2.848 | 1.251 | 18.037 | 40.132 | 0.926 |
| Internal communication is effective *(La comunicazione interna è efficace)* | 2.607 | 1.183 | 18.278 | 40.647 | 0.924 |
| In teamwork, we work well *(Si lavora bene in gruppo)* | 3.230 | 1.176 | 17.656 | 40.041 | 0.920 |
| In teamwork, we reflect on how to adapt our interventions to reach better results *(Nei gruppi di lavoro riflettiamo assieme per decidere come adattare al meglio il nostro intervento)* | 3.215 | 1.237 | 17.670 | 39.776 | 0.922 |
| Failure is a source for reflexivity, to learn from *(L'insuccesso è fonte di riflessione da cui imparare)* | 3.289 | 1.335 | 17.596 | 39.424 | 0.928 |
| The members of my organizations take part to decision making processes *(I membri della mia organizzazione prendono parte nei processi decisionali)* | 2.874 | 1.255 | 18.011 | 39.379 | 0.921 |
| The development of new ideas is promoted *(Si promuove lo sviluppo di nuove idee)* | 2.822 | 1.225 | 18.063 | 39.948 | 0.923 |

*Note.* Italian items are reported in italics and between brackets.

*Clarity about change* was measured with the three items, taken from the HSE tool indicator [52,53], aimed at capturing the clarity about the organization's change process. The higher the score, the clearer the change processes and their effects are to the employees. The items were measured with a 5-point Likert scale (1 = Never, 5 = Always). A sample item is: "When changes are made at work, I am clear how they will work out in practice". The scale showed good internal consistency (Cronbach's alpha = 0.795).

*Innovation* was measured adapting the organizational innovativeness scale by Wang and Ahmed [54], to capture the degree of organizational capability to innovate. The higher the score on the scale, the higher the capability perceived by the employees in a specific timeframe. In this study, the items referred to 2019. The scale included nine items, measured with a 5-point Likert scale (1 = not at all, 5 = yes, several). A sample item was: "[In the last year, your organization:] developed new products or services". The scale showed an excellent internal consistency (Cronbach's alpha = 0.920).

*Organizational performance* measured the perceived overall non-financial performance the organization achieved in the last year. The three-item performance scale by Putz and colleagues (2013) assessed the global rating of the quality ("We achieved remarkable results"), the quantity of work outcomes ("We managed a lot"), and the overall performance ("The performance of our organization was excellent") in the employees' eyes. The scale was measured with a 5-point Likert scale (1 = not at all, 5 = totally). It showed excellent internal consistency (Cronbach's alpha = 0.889).

### 6.2. Data Analysis

First, a confirmatory factor analysis (CFA) [55] was performed in order to examine the measurement model with MPlus version 8 [56]. To enhance our model's reliability, item parcels were created for OC (7 items) and Innovation (9 items). Two parcels defined both factors to estimate fewer free parameters and reduce sampling error sources [57–59]. Each parcel was created by sequentially summing items, assigned based on the highest to lowest item-total corrected correlations [57–59]. To deal with non-normality in the data, the robust maximum likelihood approach (MLR) was implemented [60].

Second, the structural model (Model 1, Figure 1) was tested by using the structural equation modeling (SEM) approach [55]. The model was conceptualized using OC, clarity related to change, innovation, and organizational performance. Following our hypotheses, we tested both direct and indirect (through clarity about change) effects of OC on innovation. Furthermore, we tested the clarity related to change–innovation relationship, and through it, its relationship with organizational performance.

According to a multi-faceted approach to the assessment of the fitness of the model [61], the following indices were used to evaluate the goodness-of-fit: the Chi-square likelihood ratio statistic, the Tucker and Lewis index (TLI), the comparative fit index (CFI), the root mean square error of approximation (RMSEA), with its confidence intervals, and the standardized root mean square residual (SRMR). We accepted TLI and CFI values greater than 0.95 [62], RMSEA values lower than 0.08 [63,64], and SRMR values lower than 0.08 [62,64].

## 7. Results

### 7.1. Measurement Model

The measurement model showed a good fit to the data: $\chi^2_{(30)}$ = 67.311, $p$ = 0.000, CFI = 0.976, TLI = 0.965, RMSEA = 0.068 (90% CI = 0.046–0.090, $p$ = 0.09), SRMR = 0.047, confirming validity and distinguishability of the theoretical constructs. The descriptive statistics and correlations among the studied variables are shown in Table 2. As expected, organizational performance is associated with innovation ($r$ = 0.61, $p < 0.01$), clarity about change ($r$ = 0.55, $p < 0.01$), and OC ($r$ = 0.47, $p < 0.01$). Socio-demographic (gender, age, educational level) and job-related variables (occupational conditions, contract type, weekly workload, organizational tenure) are not shown, as their associations with the variables of interest were not significant.

**Table 2.** Means, standard deviations, and correlations among organizational performance, innovation, clarity related to change, and organizational capital.

| Variables | Descriptive Statistics | | Correlations | | | |
|---|---|---|---|---|---|---|
| | **M** | **SD** | **1** | **2** | **3** | **4** |
| 1. Organizational performance | 3.29 | 0.93 | - | | | |
| 2. Innovation | 2.52 | 0.92 | 0.605 ** | - | | |
| 3. Clarity related to change | 3.15 | 1.03 | 0.551 ** | 0.440 ** | - | |
| 4. Organizational capital | 3.01 | 1.02 | 0.493 ** | 0.525 ** | 0.628 ** | - |

*Note*. M = mean, SD = standard deviation. ** = $p < 0.01$.

*7.2. Final Model*

First of all, the regression of the organizational performance on OC was verified (H1). OC showed a significant effect ($\beta$ = 0.493, $p$ < 0.01), explaining the 24.3% of organizational performance ($F_{1,268}$ = 85.976, $p$ < 0.01).

To verify the hypotheses H2, H3, H4, H5 and H6, a SEM model was performed. Such a model, hypothesizing both the direct and indirect (through clarity about change) effects of OC on innovation, as well as the direct and indirect (through innovation) effects of clarity about change on organizational performance, showed an adequate fit to the data: $\chi^2_{(30)}$ = 67.311, $p$ = 0.000, CFI = 0.976, TLI = 0.965, RMSEA = 0.068 (90% CI = 0.046–0.090, $p$ = 0.09), SRMR = 0.047. Specifically, OC was associated with Innovation ($b$ = 0.40 $p$ = 0.00) and with Clarity related to change ($b$ = 0.72, $p$ = 0.00). Thus, hypotheses H2 and H4 were confirmed. Furthermore, Clarity about change was associated with Innovation ($b$ = 0.24 $p$ = 0.01), and Innovation was related to Organizational performance ($b$ = 0.48, $p$ = 0.00). Thus, the H5 and H3a hypotheses were also confirmed. The percentages of variance explained were 34.0% for innovation, 51.9% for clarity about change, and 48.7% for organizational performance.

Innovation partially mediated the effect of OC on organizational performance ($b_{DIRECT}$ = 0.06, $p$ = 0.002; $b_{INDIRECT}$ = 0.55, $p$ = 0.000; total indirect effect: 0.09, $p$ = 0.000) [65], confirming H3b. Furthermore, innovation partially mediated the effect of clarity about change on organizational performance ($b_{DIRECT}$ = 0.36, $p$ = 0.000; $b_{INDIRECT}$ = 0.19, $p$ = 0.006; total indirect effect: 0.19, $p$ = 0.006) [65], confirming H6.

Figure 2 shows the final model.

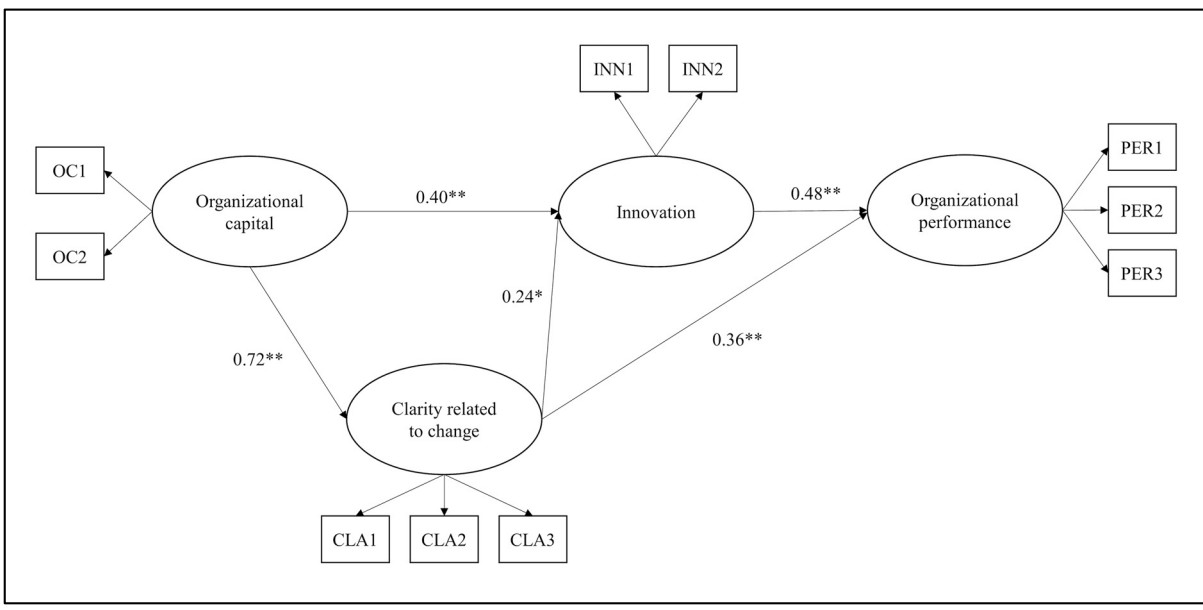

**Figure 2.** Final model. *Note.* Standardized direct effects were reported. * = $p \leq 0.05$. ** = $p \leq 0.01$.

## 8. Discussion

Our study aimed to deepen the knowledge about the relationship between the OC of PA agencies and positive organizational outcomes (namely, performance and innovation), also thanks to the role played by clarity about change. This exploratory study was developed among Italian PA agencies, an organizational sector overlooked in research on organizational capital, particularly within a performative perspective.

The proposed model confirmed our hypotheses, showing that OC was positively related to organizational performance (H1) and to organizational innovation (H2). Thus, it seems that when PA agencies successfully manage their OC, this may generate higher organizational performance, also through higher organizational innovation.

Furthermore, clarity about change played a role in the flow from OC to innovation and organizational performance. In fact, OC showed a strong association with clarity about change (H3), meaning that good communication, effective decision-making processes, reflexivity, and teamwork may affect the clarification and the sharing of organizational goals. Clarity about change also emerged as positively related to organizational innovation (H5) which, in turn, partially mediated the relationship between clarity about change and organizational performance (H6). The overall innovation, in turn, paved the way to higher performance (H3). Our model suggests that a common vision of the changes, clearly communicated and endorsed by employees, generates the alignment of efforts which, through higher innovation, was linked with higher performance. The overall model explained the 48.7% of variance.

### 8.1. Theoretical Implications

This study contributes to the literature by showing the pivotal role OC played from a performative perspective. Indeed, the opportunities offered to employees to be reflexive, to share tacit knowledge among teammates, and to take part in the decision-making process seemed to foster the organizational capability to develop new services and more tailored-made activities and thus, in general, enhancing the organization's performance. Our study focused on the public sector, which has several specificities, namely: the pursuit of both financial and nonfinancial aims, and the aim of offering services of intangible nature. This result responds to the call in the literature for deepening the knowledge about the contribution of intangible assets to organizational performance [66,67]. Thus, a first contribution of the present research was represented by the empirical evidence of the presence of the OC–performance relationship, including that among PA agencies and not only among for-profit or private organizations. Moreover, our study also confirmed the firms' innovation–organizational performance relationship [37] in the public sector. Indeed, OC proved to be a positive drive for innovation and performance among PA agencies.

This means that OC may generate organizational knowledge capable of supporting PA agencies to adapt themselves to the requests for change, and this increases the PA agencies responsiveness in modifying procedures and processes, thus being able to generate better performances to meet the needs of their customers [68]. More generally, it has to be remarked that the literature on IC, which includes the study of OC, has often neglected the public sector [10]. Thus, this research further contributes to the previous literature by exploring the dynamics behind the creation of organizational knowledge, and how they may develop added value, including among PA agencies.

The third contribution of our study relates to the psychosocial approach we adopted to define OC. To the authors' knowledge, this was the first attempt to address OC through the lens of psychosocial factors. We believe that capturing the perspective of employees' experiences both of their jobs and of the organization as a whole provides the opportunity to integrate traditional OC conceptualizations [66,67] with an understanding of the dynamics acting in the process of value creation. This psychosocial perspective is particularly important for service organizations, as is the case of PA agencies, because it captures the intangible component of the organizational factors that is also the core-nature of services. In other words, it captures the context-specific nature of OC [29].

### 8.2. Practical Implications

In addition to the theoretical aspects presented in the previous sections, the findings reached in this study suggest some practical implications for public managers. Indeed, the tested model provided first evidence of a successful pattern of management which, starting from a high level of OC, and by the means of clarity about changes, can generate innovation and then organizational performance. In other words, paying attention to the management of OC can offer an effective approach to the organizational performance and innovation of PA agencies, especially if the psychosocial dynamics are taken into account. It is important to stress the relevance of these dynamics in the light of a specific characteristic

of the PA agencies, their high level of bureaucracy, which very often may jeopardize their innovativeness and even performance [69]. Indeed, the main challenges currently faced by PA agencies regard the need to overwhelm their bureaucratic culture promoting a stronger citizen-oriented culture, [70,71].

Dramatic changes and innovation among PA agencies cannot be successfully reached only through the implementation of the various reforms that have characterized the Italian public sector. Instead, it is also necessary to implement changes in the organizational culture, in the managerial approach and in the patterns of both organizational and individual behaviors.

A recent approach to PA agencies development was inspired to the international NPM reform doctrine [72], a "through law" strategy that has shown to be only partly successful. In fact, this reform was rooted on the traditional administrative cultures and administrative regimes that has previously proven not to able to back successfully the previous reforms [73]. Our study suggested that investing on psychosocial factors of OC could be a complementary way to promote organizational changes and innovation through the enhancement of the employees' commitment.

## 9. Conclusions

Our study was carried out among Italian PA agencies. Italy, similarly to other European countries, is facing a big challenge to implement the innovation and higher performance of its public services [15] and is going through a phase of reforming the services provided. These include adapting to reduced financial and human resources, as well as changing their overall approach, from being service-oriented towards being customer-oriented. On the other hand, Italian PA agencies, as in rest of the Western world, find it difficult to fully endorse a new managerial logic, which would imply setting themselves strategic objectives linked to an increase in the level of quality, effectiveness, efficiency, and cost-effectiveness of their activities. Conversely, the management of intangible assets through the lens of IC could be key to innovate processes and pursue better performances.

Overall, our findings seem to flag the need for employees of PA agencies to feel involved in the change management process, before actively being engaged in it with their behavior. It seems that PA agencies' employees should have clear goals to pursue and know the effects that the innovation is going to generate, so that they can intentionally activate the specific behaviors aimed at achieving better performance [46,74]. More generally, our study shows that to achieve higher performance, the PAs must first use their intangible capital to facilitate the development of behaviors aimed at innovating organizational processes.

Our exploratory study has some limits that need to be highlighted. Firstly, data were collected only in one country and did not reach a statistical sample. Our findings thus cannot be generalized. It would be important to collect and compare data from different countries, in order to grasp the differences and similarities in relation to national and organizational cultures.

A second limit is that our data do not allow to understand the possible differences related to the size of the agencies reached. Other studies have outlined that organizational size may have an effect of the level of bureaucracy and the ability of an organization to innovate [75]. Furthermore, the cross-sectional design would benefit from further research testing the relationship between innovation and future performance [37], also adopting objective measures for these outcomes.

However, this study on PA agencies and OC suggests the need to further explore one's own forms of organizing work, in order to develop a capacity for reading and the critical interpretation of one's actions that allows activating new systems of thought and action than in the past. Future studies could, for instance, explore the contribution of two other intellectual capital dimensions—relational capital and human capital—to performance and innovation. Our understanding of how OC operates among PAs and contributes to their non-financial outcomes is still far from being complete, which leaves space for more studies to come.

**Author Contributions:** B.B. conceptualization, data collection, writing the original draft; I.B. data curation, analyses; M.L.F. conceptualization, writing the original draft; P.B. conceptualization, writing review, supervision. All authors have read and agreed to the published version of the manuscript.

**Funding:** This research received no external funding.

**Institutional Review Board Statement:** Ethical review and approval was not required for the study on human participants, in accordance with the local legislation and institutional requirements. The participants provided their written informed consent to participate in this study.

**Informed Consent Statement:** Informed consent was obtained from all subjects involved in the study.

**Data Availability Statement:** The raw data supporting the conclusions of this article will be made available by the authors, without undue reservation.

**Conflicts of Interest:** The authors declare no conflict of interest.

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
