# Peer review of "Organizational Capital: A Resource for Changing and Performing in Public Administrations"

_sustainability, doi:10.3390/su13105436_

Round 1

Reviewer 1 Report

The paper is interesting and as it mentions it brings a new perspective by looking at the organizational capital in the public sector and not in the private sector.

I would have some recommendations though:  

  1. Please respect the format of the journal - section should be Materials and Method. Then please try to see that the discussion section and the conclusion section are 2 distinct ones. In the discussion section you are looking into the importance / relevance and the meaning of your results. in this section you must relate your results to the ones of other researchers. In the conclusion section you must reiterate your thesis and summarize the main points of your results / evidence. This could be done in one page or half a page. I would suggest that you move the information from line 361 to the conclusions and leave the rest as discussion. 
  2. I would recommend that you explain a bit more in detail how the hypothesis were confirmed. As the paper could be of interest to general audience and not only academics I would say that a detailed explanation of the results and hypothesis validation would be good. 

Author Response

Dear Editor, dear Reviewers,

We would like to thank the three anonymous reviewers for their careful reading of our manuscript and their suportive feedback and many insightful comments. Following their suggestions, we included improvements in the manuscript (main changes are tracked in blue). Below, we will give a point-to-point replay to the comments.

Reviewer #1

The paper is interesting and as it mentions it brings a new perspective by looking at the organizational capital in the public sector and not in the private sector.

I would have some recommendations though:  

1– Please respect the format of the journal - section should be Materials and Method. Then please try to see that the discussion section and the conclusion section are 2 distinct ones. In the discussion section you are looking into the importance / relevance and the meaning of your results. in this section you must relate your results to the ones of other researchers. In the conclusion section you must reiterate your thesis and summarize the main points of your results / evidence. This could be done in one page or half a page. I would suggest that you move the information from line 361 to the conclusions and leave the rest as discussion. 

Thanks for this feedback, we have revised the entire final section both respecting the format of the Journal and integrating each Discussion and Conclusions sections in line with your and other reviewers’ thoughtful suggestions.

2– I would recommend that you explain a bit more in detail how the hypothesis were confirmed. As the paper could be of interest to general audience and not only academics I would say that a detailed explanation of the results and hypothesis validation would be good. 

We agree the discussion needed to be detailed, thus we showed results related to each hypothesis and, afretwards, we highlighted their implications for literature and applicative purposes.

Reviewer #2

Dear Authors

Your article is very interesting, and I am grateful for the opportunity to read it. I think that subject of the research is interesting, and the results of the research give a lot of new information and possibilities of further analysis.

Reading the text, I found 3 elements that I think would improve your article.

First: The abstract I think it should clearly define the purpose of the research and further analysis, the research method, the subject of research, and the results. In my opinion yours is not as clear and exact as it should be. I Think that a abstract structure is needed by the reader, but it is also important for you as authors - readers often use the abstract review method to search for content that interests them. You need to prepare more formal abstract – because it is not a story but information for the academics. If not it may be a barrier to popularizing your article.

We revised the abstract in a more structured form.

Second: There is no discussion. I think about the discussion as an attempt to confront your opinion with another and The Discussion as a chapter. So I miss both.

Thanks for this feedback, we have revised the entire final section both respecting the format of the Journal and integrating each Discussion and Conclusions sections in line with your and other reviewers’ thoughtful suggestions.

To increase the significance of the results, the discussion should embrace the differences and similarities among your findings and those of other scholars. The analysis of other studies and analyzes is clearly missing. Just as there is no reference to others points of view, different situation and examples. There is no application of a scientific analysis to reality and no real discussion. All this consequently reduces the article to the research part and causes the reader to judge the quality of the research but not its purpose or conclusions.

Thanks for this, we detailed more analitically the theoretical and practical contribution our research could imply.

Third: The Conclusions chapter is the weak part of your article. In my opinion conclusions are insufficient - The conclusion section should be a brief summary of article’s aim, methods and findings. But it's not here. This chapter should be extended. For me, the summary is too limited, there is no reference to your assumptions, your hypothesis or research questions. At this point, you should show references to your research and all formal aspects of your article. At the begging and at the end you should include a description of the research questions and research hypotheses. Develop and explain goals. It is necessary to change the convention from the presentation of research to the presentation of results and conclusions. In general, I believe that hypotheses and research questions should be presented. The goals should be presented and explained. At the end, the conclusions should refer to each of the goals.

Thanks for this, we deeply revised the final section of the paper.

Summarizing.

I find your article very good. I really like your article and appreciate your work. It is interesting topic and the conclusions could open the way for further research. You have to make some changes especially in Abstract, Discussion and Conclusions. But my general opinion and my assessment of your research and whole article is more than positive.

Reviewer #3 

Congratulations to the authors!

The article is very good.

The applied methodology is interesting.

The topic is current and interesting.

The sample used is adequate.

The hypotheses are adequate

The data is quite current (year 2020)

The academic literature is relevant.

The structure of the article is adequate.

I recommend the following for the improvement of the article:

Authors should review the bibliography. Authors should review the bibliography format. Authors should add some data that is missing from the bibliography.

For example: In reference number 1 the year is missing.

Thanks for your careful feedback, we checked all the bibliography to eliminate misprints.

Reviewer 2 Report

Dear Authors

Your article is very interesting, and I am grateful for the opportunity to read it. I think that subject of the research is interesting, and the results of the research give a lot of new information and possibilities of further analysis.

Reading the text, I found 3 elements that I think would improve your article.

First: The abstract I think it should clearly define the purpose of the research and further analysis, the research method, the subject of research, and the results. In my opinion yours is not as clear and exact as it should be. I Think that a abstract structure is needed by the reader, but it is also important for you as authors - readers often use the abstract review method to search for content that interests them. You need to prepare more formal abstract – because it is not a story but information for the academics. If not it may be a barrier to popularizing your article.

Second: There is no discussion. I think about the discussion as an attempt to confront your opinion with another and The Discussion as a chapter. So I miss both.

To increase the significance of the results, the discussion should embrace the differences and similarities among your findings and those of other scholars. The analysis of other studies and analyzes is clearly missing. Just as there is no reference to others points of view, different situation and examples. There is no application of a scientific analysis to reality and no real discussion. All this consequently reduces the article to the research part and causes the reader to judge the quality of the research but not its purpose or conclusions.

Third: The Conclusions chapter is the weak part of your article. In my opinion conclusions are insufficient - The conclusion section should be a brief summary of article’s aim, methods and findings. But it's not here. This chapter should be extended. For me, the summary is too limited, there is no reference to your assumptions, your hypothesis or research questions. At this point, you should show references to your research and all formal aspects of your article. At the begging and at the end you should include a description of the research questions and research hypotheses. Develop and explain goals. It is necessary to change the convention from the presentation of research to the presentation of results and conclusions. In general, I believe that hypotheses and research questions should be presented. The goals should be presented and explained. At the end, the conclusions should refer to each of the goals.

Summarizing.

I find your article very good. I really like your article and appreciate your work. It is interesting topic and the conclusions could open the way for further research. You have to make some changes especially in Abstract, Discussion and Conclusions. But my general opinion and my assessment of your research and whole article is more than positive.

Good luck!

Author Response

Dear Editor, dear Reviewers,

We would like to thank the three anonymous reviewers for their careful reading of our manuscript and their supportive feedback and many insightful comments. Following their suggestions, we included improvements in the manuscript (main changes are tracked in blue). Below, we will give a point-to-point replay to the comments.

Reviewer #1

The paper is interesting and as it mentions it brings a new perspective by looking at the organizational capital in the public sector and not in the private sector.

I would have some recommendations though:  

1– Please respect the format of the journal - section should be Materials and Method. Then please try to see that the discussion section and the conclusion section are 2 distinct ones. In the discussion section you are looking into the importance / relevance and the meaning of your results. in this section you must relate your results to the ones of other researchers. In the conclusion section you must reiterate your thesis and summarize the main points of your results / evidence. This could be done in one page or half a page. I would suggest that you move the information from line 361 to the conclusions and leave the rest as discussion. 

Thanks for this feedback, we have revised the entire final section both respecting the format of the Journal and integrating each Discussion and Conclusions sections in line with your and other reviewers’ thoughtful suggestions.

2– I would recommend that you explain a bit more in detail how the hypothesis were confirmed. As the paper could be of interest to general audience and not only academics I would say that a detailed explanation of the results and hypothesis validation would be good. 

We agree the discussion needed to be detailed, thus we showed results related to each hypothesis and, afterwards, we highlighted their implications for literature and applicative purposes.

Reviewer #2

Dear Authors

Your article is very interesting, and I am grateful for the opportunity to read it. I think that subject of the research is interesting, and the results of the research give a lot of new information and possibilities of further analysis.

Reading the text, I found 3 elements that I think would improve your article.

First: The abstract I think it should clearly define the purpose of the research and further analysis, the research method, the subject of research, and the results. In my opinion yours is not as clear and exact as it should be. I Think that a abstract structure is needed by the reader, but it is also important for you as authors - readers often use the abstract review method to search for content that interests them. You need to prepare more formal abstract – because it is not a story but information for the academics. If not it may be a barrier to popularizing your article.

We revised the abstract in a more structured form.

Second: There is no discussion. I think about the discussion as an attempt to confront your opinion with another and The Discussion as a chapter. So I miss both.

Thanks for this feedback, we have revised the entire final section both respecting the format of the Journal and integrating each Discussion and Conclusions sections in line with your and other reviewers’ thoughtful suggestions.

To increase the significance of the results, the discussion should embrace the differences and similarities among your findings and those of other scholars. The analysis of other studies and analyzes is clearly missing. Just as there is no reference to others points of view, different situation and examples. There is no application of a scientific analysis to reality and no real discussion. All this consequently reduces the article to the research part and causes the reader to judge the quality of the research but not its purpose or conclusions.

Thanks for this, we detailed more analitically the theoretical and practical contribution our research could imply.

Third: The Conclusions chapter is the weak part of your article. In my opinion conclusions are insufficient - The conclusion section should be a brief summary of article’s aim, methods and findings. But it's not here. This chapter should be extended. For me, the summary is too limited, there is no reference to your assumptions, your hypothesis or research questions. At this point, you should show references to your research and all formal aspects of your article. At the begging and at the end you should include a description of the research questions and research hypotheses. Develop and explain goals. It is necessary to change the convention from the presentation of research to the presentation of results and conclusions. In general, I believe that hypotheses and research questions should be presented. The goals should be presented and explained. At the end, the conclusions should refer to each of the goals.

Thanks for this, we deeply revised the final section of the paper.

Summarizing.

I find your article very good. I really like your article and appreciate your work. It is interesting topic and the conclusions could open the way for further research. You have to make some changes especially in Abstract, Discussion and Conclusions. But my general opinion and my assessment of your research and whole article is more than positive.

Reviewer #3 

Congratulations to the authors!

The article is very good.

The applied methodology is interesting.

The topic is current and interesting.

The sample used is adequate.

The hypotheses are adequate

The data is quite current (year 2020)

The academic literature is relevant.

The structure of the article is adequate.

I recommend the following for the improvement of the article:

Authors should review the bibliography. Authors should review the bibliography format. Authors should add some data that is missing from the bibliography.

For example: In reference number 1 the year is missing.

Thanks for your careful feedback, we checked all the bibliography to eliminate misprints.

Reviewer 3 Report

Congratulations to the authors!

The article is very good.

The applied methodology is interesting.

The topic is current and interesting.

The sample used is adequate.

The hypotheses are adequate

The data is quite current (year 2020)

The academic literature is relevant.

The structure of the article is adequate.

I recommend the following for the improvement of the article:

Authors should review the bibliography. Authors should review the bibliography format. Authors should add some data that is missing from the bibliography.

For example:

In reference number 1 the year is missing.

Author Response

Dear Editor, dear Reviewers,

We would like to thank the three anonymous reviewers for their careful reading of our manuscript and their supportive feedback and many insightful comments. Following their suggestions, we included improvements in the manuscript (main changes are tracked in blue). Below, we will give a point-to-point replay to the comments.

Reviewer #1

The paper is interesting and as it mentions it brings a new perspective by looking at the organizational capital in the public sector and not in the private sector.

I would have some recommendations though:  

1– Please respect the format of the journal - section should be Materials and Method. Then please try to see that the discussion section and the conclusion section are 2 distinct ones. In the discussion section you are looking into the importance / relevance and the meaning of your results. in this section you must relate your results to the ones of other researchers. In the conclusion section you must reiterate your thesis and summarize the main points of your results / evidence. This could be done in one page or half a page. I would suggest that you move the information from line 361 to the conclusions and leave the rest as discussion. 

Thanks for this feedback, we have revised the entire final section both respecting the format of the Journal and integrating each Discussion and Conclusions sections in line with your and other reviewers’ thoughtful suggestions.

2– I would recommend that you explain a bit more in detail how the hypothesis were confirmed. As the paper could be of interest to general audience and not only academics I would say that a detailed explanation of the results and hypothesis validation would be good. 

We agree the discussion needed to be detailed, thus we showed results related to each hypothesis and, afretwards, we highlighted their implications for literature and applicative purposes.

Reviewer #2

Dear Authors

Your article is very interesting, and I am grateful for the opportunity to read it. I think that subject of the research is interesting, and the results of the research give a lot of new information and possibilities of further analysis.

Reading the text, I found 3 elements that I think would improve your article.

First: The abstract I think it should clearly define the purpose of the research and further analysis, the research method, the subject of research, and the results. In my opinion yours is not as clear and exact as it should be. I Think that a abstract structure is needed by the reader, but it is also important for you as authors - readers often use the abstract review method to search for content that interests them. You need to prepare more formal abstract – because it is not a story but information for the academics. If not it may be a barrier to popularizing your article.

We revised the abstract in a more structured form.

Second: There is no discussion. I think about the discussion as an attempt to confront your opinion with another and The Discussion as a chapter. So I miss both.

Thanks for this feedback, we have revised the entire final section both respecting the format of the Journal and integrating each Discussion and Conclusions sections in line with your and other reviewers’ thoughtful suggestions.

To increase the significance of the results, the discussion should embrace the differences and similarities among your findings and those of other scholars. The analysis of other studies and analyzes is clearly missing. Just as there is no reference to others points of view, different situation and examples. There is no application of a scientific analysis to reality and no real discussion. All this consequently reduces the article to the research part and causes the reader to judge the quality of the research but not its purpose or conclusions.

Thanks for this, we detailed more analitically the theoretical and practical contribution our research could imply.

Third: The Conclusions chapter is the weak part of your article. In my opinion conclusions are insufficient - The conclusion section should be a brief summary of article’s aim, methods and findings. But it's not here. This chapter should be extended. For me, the summary is too limited, there is no reference to your assumptions, your hypothesis or research questions. At this point, you should show references to your research and all formal aspects of your article. At the begging and at the end you should include a description of the research questions and research hypotheses. Develop and explain goals. It is necessary to change the convention from the presentation of research to the presentation of results and conclusions. In general, I believe that hypotheses and research questions should be presented. The goals should be presented and explained. At the end, the conclusions should refer to each of the goals.

Thanks for this, we deeply revised the final section of the paper.

Summarizing.

I find your article very good. I really like your article and appreciate your work. It is interesting topic and the conclusions could open the way for further research. You have to make some changes especially in Abstract, Discussion and Conclusions. But my general opinion and my assessment of your research and whole article is more than positive.

Reviewer #3 

Congratulations to the authors!

The article is very good.

The applied methodology is interesting.

The topic is current and interesting.

The sample used is adequate.

The hypotheses are adequate

The data is quite current (year 2020)

The academic literature is relevant.

The structure of the article is adequate.

I recommend the following for the improvement of the article:

Authors should review the bibliography. Authors should review the bibliography format. Authors should add some data that is missing from the bibliography.

For example: In reference number 1 the year is missing.

Thanks for your careful feedback, we checked all the bibliography to eliminate misprints.